# Posterior Inference with Diffusion Models
# for High-dimensional Black-box Optimization

Taeyoung Yun [* 1]   Kiyoung Om [* 1]   Jaewoo Lee [1]   Sujin Yun [1]   Jinkyoo Park [1]

## Abstract

Optimizing high-dimensional and complex black-box functions is crucial in numerous scientific applications. While Bayesian optimization (BO) is a powerful method for sample-efficient optimization, it struggles with the curse of dimensionality and scaling to thousands of evaluations. Recently, leveraging generative models to solve black-box optimization problems has emerged as a promising framework. However, those methods often underperform compared to BO methods due to limited expressivity and difficulty of uncertainty estimation in high-dimensional spaces. To overcome these issues, we introduce **DiBO**, a novel framework for solving high-dimensional black-box optimization problems. Our method iterates two stages. First, we train a diffusion model to capture the data distribution and deep ensembles to predict function values with uncertainty quantification. Second, we cast the candidate selection as a posterior inference problem to balance exploration and exploitation in high-dimensional spaces. Concretely, we fine-tune diffusion models to amortize posterior inference. Extensive experiments demonstrate that our method outperforms state-of-the-art baselines across synthetic and real-world tasks. Our code is publicly available here.

## 1. Introduction

Optimizing high-dimensional and complex black-box functions is crucial in various scientific and engineering applications, including hyperparameter optimization (Šehić et al., 2022), material discovery (Hernández-Lobato et al., 2017), chemical design (Gómez-Bombarelli et al., 2018; Griffiths et al., 2023), and control systems (Candelieri et al., 2018).

Bayesian optimization (BO) (Kushner, 1964; Garnett, 2023) is a powerful method for solving black-box optimization. BO constructs a surrogate model from observed data and finds an input that maximizes the acquisition function to query the black-box function. However, it scales poorly to high dimensions due to the curse of dimensionality (Wang et al., 2016) and struggles scaling to thousands of evaluations (Wang et al., 2018; Eriksson et al., 2019). To address these challenges, several approaches have been suggested to scale up BO for high-dimensional optimization problems. Some works propose a mapping from high-dimensional space into low-dimensional subspace (Gómez-Bombarelli et al., 2018; Maus et al., 2022; Nayebi et al., 2019; Letham et al., 2020) or assume additive structures of the target function (Duvenaud et al., 2011; Rolland et al., 2018) to perform optimization in low-dimensional spaces. However, these methods often rely on unrealistic assumptions.

Other works partition the search space into promising local regions and search candidates within such regions (Eriksson et al., 2019; Wang et al., 2020), which exhibits promising results. However, these methods may struggle to escape from local optima within a limited number of evaluations due to several challenges. First, it is notoriously difficult to find an input that maximizes the acquisition function in high-dimensional spaces since the function is often highly non-convex and contains numerous local optima (Ament et al., 2023). Furthermore, as the data points lie on a tiny manifold compared to the entire search space, the uncertainty becomes extremely high in regions too far from the dataset (Oh et al., 2018).

Recently, generative model-based approaches have shown promising results in black-box optimization (Brookes et al., 2019; Kumar & Levine, 2020; Wu et al., 2024). These methods utilize an inverse mapping from function values to the input domain and propose candidates via sampling from the trained model conditioned on a high score. While they alleviate aforementioned issues in BO by converting optimization as sampling (Janner et al., 2022), their performance degrades in higher dimensions due to the limited expressivity of underlying models (Brookes et al., 2019; Kumar & Levine, 2020) or the difficulty of uncertainty estimation in high-dimensional spaces (Wu et al., 2024).

---

[*]Equal contribution   [1]Korea Advanced Institute of Science and Technology (KAIST). Correspondence to: Taeyoung Yun <99yty@kaist.ac.kr>, Kiyoung Om <se99an@kaist.ac.kr>.

*Proceedings of the 42nd International Conference on Machine Learning*, Vancouver, Canada. PMLR 267, 2025. Copyright 2025 by the author(s).

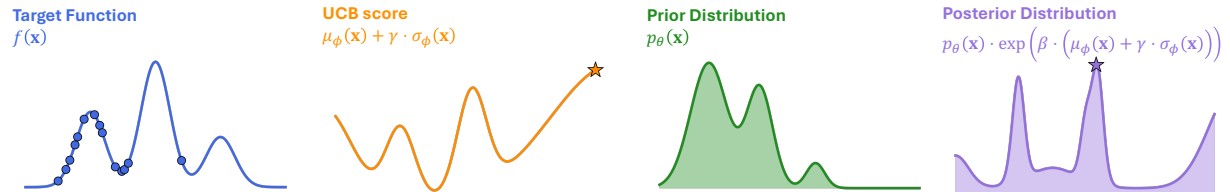

*Figure 1.* Motivating example of our method. In high-dimensional spaces, directly searching for input that maximizes UCB score may lead to sub-optimal results (As depicted in ⭐ of the second figure). Sampling from the posterior distribution prevents overemphasized exploration in the boundary of search space and leads to efficient exploration (As depicted in ⭐ of the last figure).

To overcome these issues, we introduce **Di**ffusion models for **B**lack-box **O**ptimization (**DiBO**), a novel generative model-based approach for high-dimensional black-box optimization. Our key idea is to cast the candidate selection problem as a posterior inference problem. Specifically, we first train a proxy and diffusion model, which serves as a reward function and the prior. Then, we construct a posterior distribution by multiplying two components and sample candidates from the posterior to balance exploration and exploitation in high-dimensional spaces. Instead of searching for input that maximizes the acquisition function, sampling from the posterior prevents us from choosing the samples that lie too far from the dataset, as illustrated in Figure 1.

Our method iterates two stages. First, we train a diffusion model to effectively capture high-dimensional data distribution and an ensemble of proxies to predict function values with uncertainty quantification. During training, we adopt a reweighted training scheme proposed in prior generative model-based approaches to focus on high-scoring data points (Kumar & Levine, 2020; Tripp et al., 2020). Second, we sample candidates from the posterior distribution. As the sampling from the posterior distribution is intractable, we train an amortized sampler to generate unbiased samples from the posterior by fine-tuning the diffusion model. While the posterior distribution remains highly non-convex, we can capture such complex and multi-modal distribution by exploiting the expressivity of the diffusion models. By repeating these two stages, we progressively get close to the high-scoring regions of the target function.

As we have a limited budget for evaluations, it is beneficial to choose modes of the posterior distribution as proposing candidates. To accomplish this, we propose two post-processing strategies after training the amortized sampler: local search and filtering. Concretely, we generate many samples from the amortized sampler and improve them via local search. Then, we filter candidates with respect to the unnormalized posterior density. By incorporating these strategies, we can further boost the sample efficiency of our method across various optimization tasks.

We conduct extensive experiments on four synthetic and three real-world high-dimensional black-box optimization tasks. We demonstrate that our method achieves superior performance on a variety of tasks compared to state-of-the-art baselines, including BO methods, generative model-based methods, and evolutionary algorithms.

## 2. Preliminaries

### 2.1. Black-box Optimization

In black-box optimization, our objective is to find a design $\mathbf{x} \in \mathcal{X}$ that maximizes the target black-box function $f(\mathbf{x})$.

$$\text{find } \mathbf{x}^* = \arg\max_{\mathbf{x} \in \mathcal{X}} f(\mathbf{x})$$

$$\text{with } R \text{ rounds of } B \text{ batch of queries} \quad (1)$$

Querying designs in batches is practical in many real-world scenarios, such as biological sequence designs (Jain et al., 2022; Kim et al., 2024a). As the evaluation process is expensive in most cases, developing an algorithm with high sample efficiency is critical in black-box optimization.

### 2.2. Diffusion Probabilistic Models

Diffusion probabilistic models (Sohl-Dickstein et al., 2015; Ho et al., 2020) are a class of generative models that aim to approximate the true distribution $q_0(\mathbf{x}_0)$ with a parametrized model of the form: $p_\theta(\mathbf{x}_0) = \int p_\theta(\mathbf{x}_{0:T}) \, d\mathbf{x}_{1:T}$, where $\mathbf{x}_0$ and latent variables $\mathbf{x}_1, \cdots, \mathbf{x}_T$ share the same dimensionality. The joint distribution $p_\theta(\mathbf{x}_{0:T})$, often referred to as the *reverse process*, is defined via a Markov chain that starts from a standard Gaussian prior $p_T(\mathbf{x}_T) = \mathcal{N}(\mathbf{0}, \mathbf{I})$:

$$p_\theta(\mathbf{x}_{0:T}) = p_T(\mathbf{x}_T) \prod_{t=1}^{T} p_\theta(\mathbf{x}_{t-1}|\mathbf{x}_t), \quad (2)$$

$$p_\theta(\mathbf{x}_{t-1}|\mathbf{x}_t) = \mathcal{N}\big(\mu_\theta(\mathbf{x}_t, t), \boldsymbol{\Sigma}_t\big). \quad (3)$$

$p_\theta(\mathbf{x}_{t-1}|\mathbf{x}_t)$ is a Gaussian transition from step $t$ to $t-1$.

We choose *forward process* as fixed to be a Markov chain that progressively adds Gaussian noise to the data according to a variance schedule $\beta_1, \ldots, \beta_T$:

$$q(\mathbf{x}_{1:T}|\mathbf{x}_0) = \prod_{t=1}^{T} q\big(\mathbf{x}_t|\mathbf{x}_{t-1}\big), \quad (4)$$

$$q\big(\mathbf{x}_t|\mathbf{x}_{t-1}\big) = \mathcal{N}\big(\sqrt{1-\beta_t}\,\mathbf{x}_{t-1}, \beta_t\mathbf{I}\big). \quad (5)$$

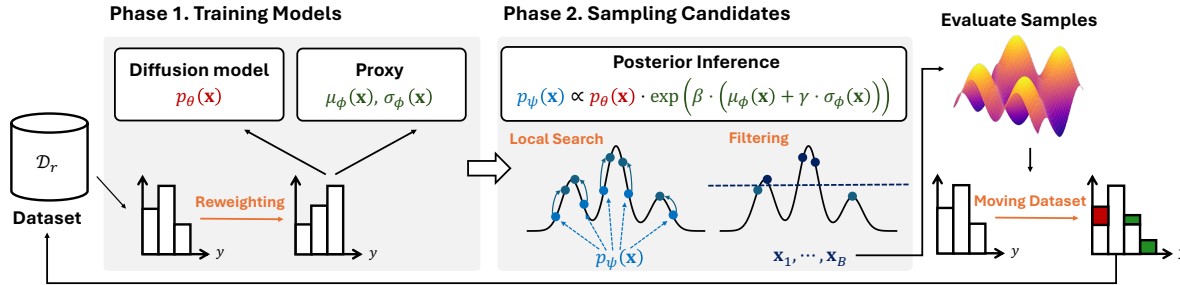

*Figure 2.* Overview of our method. **Phase 1:** Train diffusion models and ensemble of proxies. **Phase 2:** Sampling candidates from the posterior distribution and post-processing via local search and filtering. Then, we evaluate samples, update the dataset, and repeat the process until we find an optimal design.

**Training Diffusion Models.** We can train diffusion models by optimizing the variational lower bound of negative log-likelihood, $\mathbb{E}_{q_0}[-\log p_\theta(\mathbf{x}_0)]$. Following Ho et al. (2020), we use a simplified loss with noise parameterization:

$$\mathcal{L}(\theta) = \mathbb{E}_{\mathbf{x}_0 \sim q_0,\, t \sim U(1,T),\, \epsilon \sim \mathcal{N}(0,I)} \big[ \|\epsilon - \epsilon_\theta(\mathbf{x}_t, t)\|^2 \big] \quad (6)$$

Here, $\epsilon_\theta(\mathbf{x}_t, t)$ is the learned noise estimator, and

$$\mu_\theta(\mathbf{x}_t, t) = \frac{1}{\sqrt{\alpha_t}} \Big( \mathbf{x}_t - \frac{\sqrt{\beta_t}}{\sqrt{1 - \bar{\alpha}_t}} \epsilon_\theta(\mathbf{x}_t, t) \Big), \quad (7)$$

where $\alpha_t = 1 - \beta_t$ and $\bar{\alpha}_t = \prod_{s=1}^{t} \alpha_s$.

**Fine-tuning Diffusion Models for Posterior Inference.**
Given a diffusion model $p_\theta(\mathbf{x})$ and a reward function $r(\mathbf{x})$, we can define a posterior distribution $p^{\text{post}}(\mathbf{x}) \propto p_\theta(\mathbf{x})r(\mathbf{x})$, where the diffusion model serves as a prior. Sampling from the posterior distribution enables us to solve various downstream tasks (Chung et al., 2023; Lu et al., 2023; Venkatraman et al., 2024). For example, in offline reinforcement learning, if we train a conditional diffusion model $\mu(a|s)$ as a behavior policy, it requires sampling from the product distribution of behavior policy and Q-function (Nair et al., 2020), i.e., $\pi(a|s) \propto \mu(a|s) \exp(\beta \cdot Q(s,a))$.

When the prior is modeled as a diffusion model, the sampling from the posterior distribution is intractable due to the hierarchical nature of the sampling process. Fortunately, we can utilize relative trajectory balance (RTB) loss suggested by Venkatraman et al. (2024) to learn amortized sampler $p_\psi$ that approximates the posterior distribution by fine-tuning the prior diffusion model as follows:

$$\mathcal{L}(\mathbf{x}_{0:T}; \psi) = \left( \log \frac{Z_\psi \cdot p_\psi(\mathbf{x}_{0:T})}{r(\mathbf{x}_0) \cdot p_\theta(\mathbf{x}_{0:T})} \right)^2 \quad (8)$$

where $\mathbf{x}_0 = \mathbf{x}$ and $Z_\psi$ is the partition function estimator. If the loss converges to zero for all possible trajectories $\mathbf{x}_{0:T}$, the amortized sampler matches the posterior distribution.

One of the main advantages of RTB loss is that we can train the model in an off-policy manner. In other words, we

can train the model with the samples from the distribution different from the current policy $p_\psi$ to ensure mode coverage (Sendera et al., 2024; Akhound-Sadegh et al., 2024).

## 3. Method

In this section, we introduce **DiBO**, a novel approach for high-dimensional black-box optimization by leveraging diffusion models. Our method iterates two stages to find an optimal design in high-dimensional spaces. First, we train a diffusion model to capture the data distribution and an ensemble of proxies to predict function values with uncertainty quantification. During training, we apply a reweighted training scheme to focus on high-scoring regions. Next, we sample candidates from the posterior distribution. To further improve sample efficiency, we adopt local search and filtering to select diverse modes of posterior distribution as candidates. We then evaluate the selected candidates, update the dataset, and repeat the process until we find an optimal design. Figure 2 shows the overview of our method.

### 3.1. Phase 1: Training Models

In each round $r$, we have a pre-collected dataset of input-output pairs $\mathcal{D}_r = \{(\mathbf{x}_i, y_i)\}_{i=1}^{I}$, where $I$ is the number of data points.

**Training Diffusion Model.** We first train a diffusion model $p_\theta(\mathbf{x})$ using $\mathcal{D}_r$ to capture the data distribution. We choose a diffusion model as it has a powerful capability to learn the distribution of high-dimensional data across various domains (Ramesh et al., 2022; Ho et al., 2022).

**Training Ensemble of Proxies.** We also train a proxy model to predict function values using the dataset $\mathcal{D}_r$. As it is notoriously difficult to accurately predict all possible regions in high-dimensional spaces with a limited amount of samples, we need to properly quantify the uncertainty of our proxy model. To this end, we train an ensemble of proxies $f_{\phi_1}, \cdots, f_{\phi_K}$ to estimate the epistemic uncertainty of the model (Lakshminarayanan et al., 2017).

**Reweighted Training.** In the training stage, we introduce a reweighted training scheme to focus on high-scoring data points since our objective is to find an optimal design that maximizes the target black-box function. Reweighted training has been widely used in generative modeling for black-box optimization, especially in offline settings (Kumar & Levine, 2020; Krishnamoorthy et al., 2023; Kim et al., 2024b). Formally, we can compute the weight for each data point as follows:

$$w(y, \mathcal{D}_r) = \frac{\exp(y)}{\sum_{(\mathbf{x}', y') \in \mathcal{D}_r} \exp(y')} \quad (9)$$

Then, our training objective for proxies and diffusion models can be described as follows:

$$\mathcal{L}(\phi_{1:K}) = \sum_{k=1}^{K} \sum_{(\mathbf{x}, y) \in \mathcal{D}_r} w(y, \mathcal{D}_r) \left(y - f_{\phi_k}(\mathbf{x})\right)^2 \quad (10)$$

$$\mathcal{L}(\theta) = - \sum_{(\mathbf{x}, y) \in \mathcal{D}_r} w(y, \mathcal{D}_r) \log p_\theta(\mathbf{x}). \quad (11)$$

### 3.2. Phase 2: Sampling Candidates

After training models, we sample candidates from the posterior distribution to query the black-box function. As sampling from the posterior is intractable, we introduce an amortized sampler that generates unbiased samples from the posterior by fine-tuning the pre-trained diffusion model. Then, to further boost the sample efficiency, we apply post-processing strategies, local search and filtering, to select diverse modes of the posterior distribution as candidates.

**Amortizing Posterior Inference.** Our key idea is to sample candidates from the probability distribution that satisfies two desiderata: (1) promote exploration towards both high-rewarding and highly uncertain regions and (2) prevent the sampled candidates from deviating excessively from the data distribution. To accomplish these objectives, we can define our target distribution as follows:

$$p_{\text{tar}}(\mathbf{x}) = \arg\max_{p \in \mathcal{P}} \mathbb{E}_{\mathbf{x} \sim p} \left[r_\phi(\mathbf{x})\right] - \frac{1}{\beta} \cdot D_{\text{KL}} \left(p \parallel p_\theta\right) \quad (12)$$

where $r_\phi(\mathbf{x}) = \mu_\phi(\mathbf{x}) + \gamma \cdot \sigma_\phi(\mathbf{x})$ is UCB score from the proxy, and $\mu_\phi(\mathbf{x}), \sigma_\phi(\mathbf{x})$ indicate mean and the standard deviation from proxy predictions, respectively. $\mathcal{P}$ denotes the set of feasible probability distributions and $\beta$ is an inverse temperature. Target distribution that maximizes the right part of the Equation (12) can be analytically derived as follows (Nair et al., 2020):

$$p_{\text{tar}}(\mathbf{x}) = \frac{1}{Z} \cdot p_\theta(\mathbf{x}) \exp\left(\beta \cdot r_\phi(\mathbf{x})\right) \quad (13)$$

where $Z = \int_{\mathbf{x} \in \mathcal{X}} p_\theta(\mathbf{x}) \exp\left(\beta \cdot r_\phi(\mathbf{x})\right)$ is a partition function. As we do not know the partition function, sampling

---

**Algorithm 1** DiBO

1: **Input:** Initial dataset $\mathcal{D}_0$; Max rounds $R$; Batch size $B$; Buffer size $L$; Diffusion model $p_\theta$, $p_\psi$; Proxy $f_{\phi_1}, \cdots f_{\phi_K}$
2: **for** $r = 0, \ldots, R - 1$ **do**
3:     **Phase 1. Training Models**
4:     Compute weights $w(y, \mathcal{D}_r)$ with Equation (9)
5:     Train $f_{\phi_1}, \cdots f_{\phi_K}$ with Equation (10)
6:     Train $p_\theta$ with Equation (11)
7:
8:     **Phase 2. Sampling Candidates**
9:     Initialize $p_\psi \leftarrow p_\theta$
10:     Train $p_\psi$ with Equation (14) using $\mathbf{x}_{0:T}$ from $p_\psi$ or from the dataset $\mathcal{D}_r$
11:     Sample $\{\mathbf{x}_i\}_{i=1}^M \sim p_\psi(\mathbf{x})$
12:     Update $\{\mathbf{x}_i\}_{i=1}^M$ into $\{\mathbf{x}_i^*\}_{i=1}^M$ with Equation (15)
13:     Filter top-$B$ samples $\{\mathbf{x}_b\}_{b=1}^B$ among $\{\mathbf{x}_i^*\}_{i=1}^M$
14:
15:     **Evaluation and Moving Dataset**
16:     Evaluate $y_b = f(\mathbf{x}_b), \quad \forall b = 1, \cdots, B$
17:     Update $\mathcal{D}_{r+1} \leftarrow \mathcal{D}_r \cup \left\{(\mathbf{x}_b, y_b)\right\}_{b=1}^B$
18:     **if** $|\mathcal{D}_{r+1}| > L$ **then**
19:         Remove bottom-$(|\mathcal{D}_{r+1}| - L)$ samples from $\mathcal{D}_{r+1}$
20:     **end if**
21: **end for**

---

from $p_{\text{tar}}$ is intractable. Therefore, we introduce amortized sampler $p_\psi \approx p_{\text{tar}}$, which can be obtained by fine-tuning the trained diffusion model with relative trajectory balance loss suggested by Venkatraman et al. (2024). Formally, we train the parameters of $p_\psi$ with the following objective:

$$\mathcal{L}(\mathbf{x}_{0:T}; \psi) = \left(\log \frac{Z_\psi \cdot p_\psi(\mathbf{x}_{0:T})}{\exp\left(\beta \cdot r_\phi(\mathbf{x}_0)\right) \cdot p_\theta(\mathbf{x}_{0:T})}\right)^2 \quad (14)$$

where $\mathbf{x}_0 = \mathbf{x}$, and $Z_\psi$ is a parameterized partition function.

As mentioned in the previous section, we can employ off-policy training to effectively match the target distribution. To this end, we train $p_\psi$ with the on-policy trajectories from the model mixed with the trajectories generated by samples from the pre-collected dataset $\mathcal{D}_r$. Please refer to Appendix B.2.1 for more details on off-policy training.

After training $p_\psi$, we can generate unbiased samples from our target distribution. However, as we have a limited evaluation budget, it is advantageous to refine candidates to exhibit a higher probability density of target distribution, i.e., modes of distribution. To achieve this, we introduce two post-processing strategies: local search and filtering.

**Local Search.** First, we generate a set of candidates $\{\mathbf{x}_i\}_{i=1}^M$ by sampling from $p_\psi$. For each candidate $\mathbf{x}_i$, we perform a local search using gradient ascent to move it toward high-density regions. Formally, we update the original

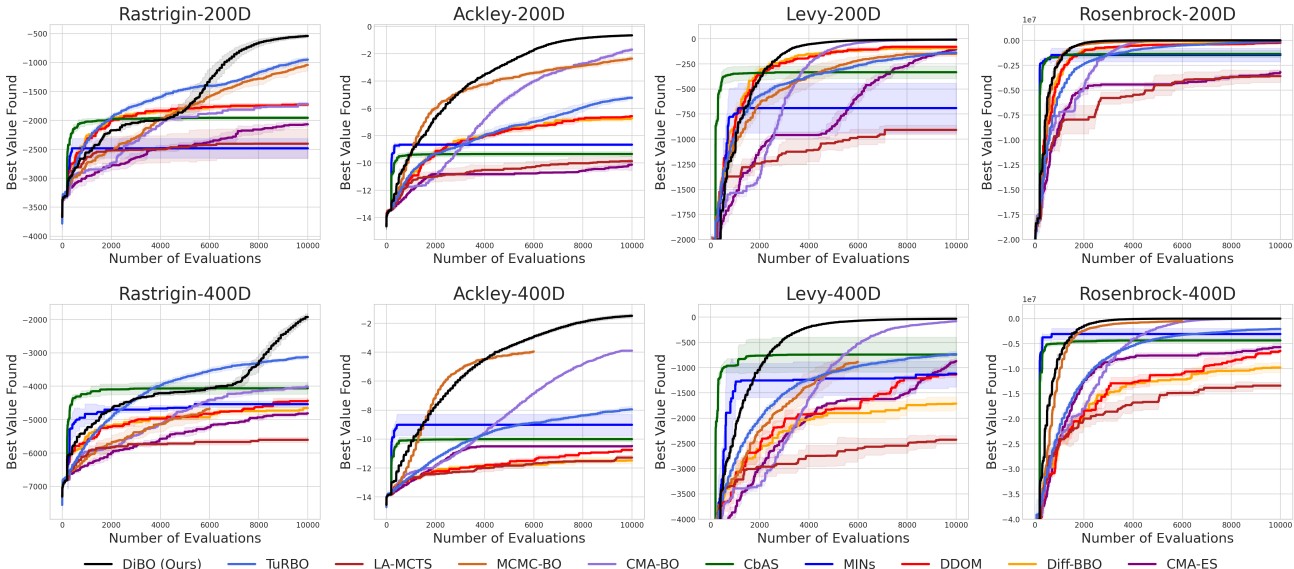

*Figure 3.* Comparison between our method against baselines in synthetic tasks. Experiments are conducted with four random seeds, and the mean and one standard deviation are reported.

candidate $\mathbf{x}_i$ to $\mathbf{x}_i^*$ as follows:

$$\mathbf{x}_i^{j+1} \leftarrow \mathbf{x}_i^j + \eta \cdot \nabla_{\mathbf{x}=\mathbf{x}_i^j} \left( p_\theta(\mathbf{x}) \cdot \exp(\beta \cdot r_\phi(\mathbf{x})) \right), \quad (15)$$

$$\text{for } j = 0, \ldots, J-1, \quad \text{where } \mathbf{x}_i^0 = \mathbf{x}_i, \ \mathbf{x}_i^J = \mathbf{x}_i^* \quad (16)$$

where $\eta$ is the step size, and $J$ is the number of updates. To estimate the marginal probability $p_\theta(\mathbf{x})$, we employ the probability flow ordinary differential equation (PF ODE) Song et al. (2021) with a differentiable ODE solver. Please refer to Appendix B.2.2 for more details on local search.

As the amortized sampler $p_\psi$ is parametrized as a diffusion model, we can expect that it generates samples across diverse possible promising regions. Then, the local search procedure guides samples towards modes of each promising region (Kim et al., 2024c).

**Filtering.** After the local search, we introduce filtering to select $B$ candidates for evaluation among generated samples $\{\mathbf{x}_1^*, \cdots, \mathbf{x}_M^*\}$. To be specific, we select the top-$B$ samples with respect to the unnormalized target density, $p_\theta(\mathbf{x}) \cdot \exp(\beta \cdot r_\phi(\mathbf{x}))$. Through filtering, we can effectively capture the high-quality modes of the posterior distribution, thereby significantly improving the sample efficiency.

### 3.3. Evaluation and Moving Dataset

After selecting candidates, we evaluate their function values by querying the black-box function. Then, we update the dataset with new observations. When updating the dataset, we remove the samples with the lowest function values if the size of the dataset is larger than the buffer size $L$. We empirically find that it reduces the computational complexity during training and ensures that the model concentrates more on the high-scoring regions.

## 4. Experiments

In this section, we present experimental results on high-dimensional black-box optimization tasks. First, we conduct experiments on four synthetic functions commonly used in the BO literature (Eriksson et al., 2019; Wang et al., 2020). Then, we perform experiments on three high-dimensional real-world tasks, including HalfCheetah-102D from Mu-JoCo Locomotion, RoverPlanning-100D, and DNA-180D from LassoBench. The description of each task is available in Appendix A.

### 4.1. Baselines

We evaluate our method against state-of-the-art (SOTA) baselines for high-dimensional black-box optimization, including BO methods: TuRBO (Eriksson et al., 2019), LA-MCTS (Wang et al., 2020), MCMC-BO (Yi et al., 2024), CMA-BO (Ngo et al., 2024), an evolutionary search approach: CMA-ES (Hansen, 2006), and existing generative model-based algorithms: CbAS (Brookes et al., 2019), MINs (Kumar & Levine, 2020), DDOM (Krishnamoorthy et al., 2023), and Diff-BBO (Wu et al., 2024). Details about baseline implementation can be found in Appendix C.

### 4.2. Synthetic function tasks

We benchmark four synthetic functions that are widely used for evaluating high-dimensional black-box optimization algorithms: *Rastrigin, Ackley, Levy,* and *Rosenbrock.* We evaluate each function in both $D = 200$ and $400$ dimensions and set the search space $\mathcal{X} = [\text{lb}, \text{ub}]^D$ for each function following previous works (Wang et al., 2020; Yi et al., 2024). All experiments are conducted with initial dataset

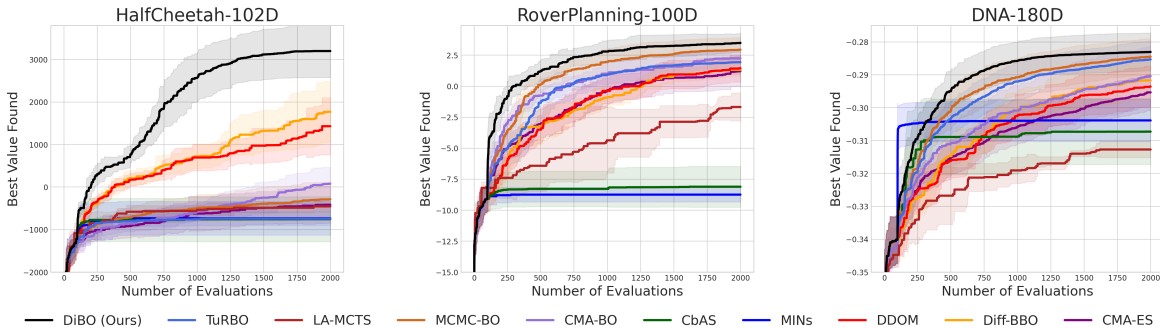

*Figure 4.* Comparison between our method against baselines in real-world tasks. Experiments are conducted with ten random seeds, and the mean and one standard deviation are reported.

size $|\mathcal{D}_0| = 200$, batch size $B = 100$, and $10,000$ as the maximum evaluation limit.[1]

As shown in Figure 3, our method significantly outperforms all baselines across four synthetic functions. Furthermore, we observe that our method not only discovers high-scoring designs but also achieves high sample efficiency. Specifically, our method exhibits a significant gap compared to baselines on *Rastrigin* and *Ackley* tasks, which have numerous local optima near the global optimum. It highlights that our key idea, sampling candidates from the posterior distribution, enables effective exploration of promising regions in high-dimensional spaces and mitigates the risk of converging sub-optimal regions.

We find that generative model-based approaches such as CbAS and MINs perform well in the early stage but struggle to improve the performance through subsequent iterations. While diffusion-based methods (DDOM, Diff-BBO) show consistent improvements across various tasks, the performance lags behind that of recent BO methods. These results demonstrate that the superiority of our method stems not just from using diffusion models but also from our novel framework calibrated for high-dimensional black-box optimization. We also conduct an ablation study on each component of our method in the following section.

We also compare our method with high-dimensional Bayesian optimization methods. While these approaches exhibit comparable performance and often outperform generative model-based baselines, they remain relatively sample-inefficient compared to our method. It reveals that sampling diverse candidates from the posterior distribution can be a sample-efficient solution for high-dimensional black-box optimization compared to choosing inputs that maximize the acquisition function.

### 4.3. Real-World Tasks

To evaluate the performance and adaptability of our method in real-world scenarios, we conduct experiments on three additional tasks: HalfCheetah-102D, RoverPlanning-100D,

and DNA-180D. Each experiment starts with $|\mathcal{D}_0| = 100$ initial samples, a batch size of $B = 50$, and a maximum evaluation limit of $2,000$.

The results are illustrated in Figure 4. Our method exhibits superiority in terms of both the performance and the sample efficiency compared to baseline approaches. While other methods show inconsistent performance—excelling on some tasks but underperforming on others—our method consistently surpassed the baselines, highlighting its robustness across a broader range of tasks.

## 5. Additional Analysis

In this section, we carefully analyze the effectiveness of each component of our method. We conduct additional analysis in Rastrigin-200D and HalfCheetah-102D tasks.

**Ablation on Reweighted Training.** We examine the impact of reweighted training during the model training stage. We conduct experiments by omitting the reweighting component. As shown in Figure 5a, the performance of our method significantly drops if we remove the reweighted training. It underscores that focusing on high-scoring regions accelerates the optimization process. We also conduct analysis on the number of training epochs in Appendix D.1. We find that naively increasing the number of training epochs does not lead to an improvement in performance.

**Ablation on Sampling Procedure.** We analyze the effect of strategies we have proposed during the sampling stage. We conduct experiments without filtering, local search, and finally completely remove the amortized inference stage and propose samples from the diffusion model $p_\theta$.

As depicted in Figure 5b, each component of our method significantly affects the performance. Notably, when we remove both local search and filtering strategies, we observe that the sample efficiency of our method significantly drops,

---

[1]For MCMC-BO, we report the score of budget $6,000$ on tasks with $D = 400$ due to memory constraints.

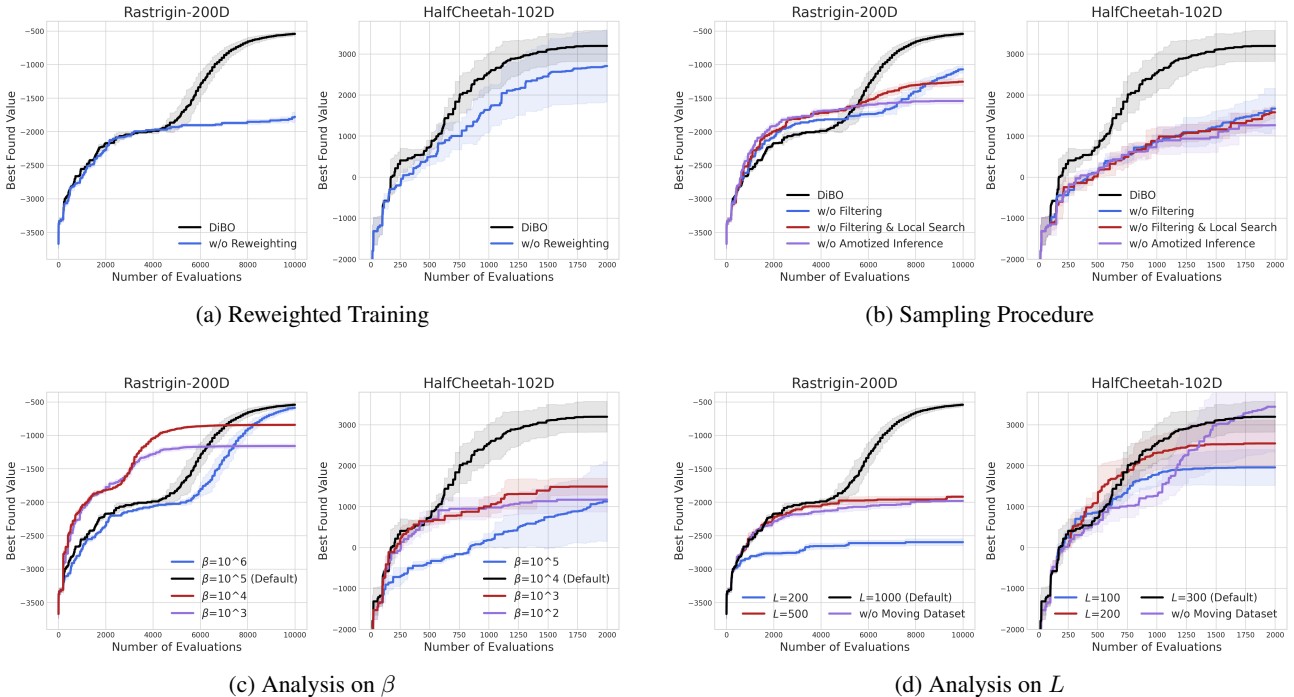

(a) Reweighted Training

(b) Sampling Procedure

(c) Analysis on $\beta$

(d) Analysis on $L$

*Figure 5.* Ablation on various components in DiBO. Experiments are conducted on the Rastrigin-200D and HalfCheetah-102D tasks.

demonstrating the effectiveness of the proposed components. We also conduct further analysis on the number of local search steps $J$ in Appendix D.2 and the effect of off-policy training for amortized inference in Appendix D.3.

**Analysis on Inverse Temperature $\beta$.** The parameter $\beta$ in Equation (12) governs the trade-off between exploitation and exploration. If $\beta$ is too small, the method tends to exploit already discovered high-scoring regions. On the other hand, if $\beta$ is too large, we generate samples that deviate too far from the current dataset and overemphasize exploration of the boundary of search space.

As shown in Figure 5c, when $\beta$ is too small, it often leads to convergence on a local optimum due to limited exploration. Conversely, if $\beta$ is too large, the model becomes overly dependent on the proxy function, resulting in excessive exploration and ultimately slowing the convergence.

**Analysis on Buffer size $L$.** As we described in Section 3.3, we introduce buffer size $L$ to maintain the dataset with high-scoring samples collected during the evaluation cycles. The choice of $L$ impacts both the time complexity and the sample efficiency of our method.

As illustrated in Figure 5d, using a small buffer size results in reaching suboptimal results, also causing early performance saturation. In contrast, a larger buffer size can reach the optimal value as in our default setting but significantly decelerate the rate of performance improvement.

**Analysis on initial dataset size $|\mathcal{D}_0|$ and batch size $B$.** Initial dataset size and batch size for each round can be crucial in the performance of the method. To verify the effect of these components, we conduct additional analysis in Appendices D.4 and D.5 and find that our method is robust to different initial experiment settings.

**Analysis on uncertainty estimation.** We further analyze how the uncertainty estimation affects the performance of our method in Appendix D.6. We find that using ensembles to measure uncertainty is a powerful approach, and using the UCB score based on the uncertainty can effectively guide the model to explore promising regions.

**Time complexity of the method.** We conduct analysis on the time complexity of our method in Appendix D.7. Our findings show that our method exhibits a relatively low or comparable running time compared to other baselines.

# 6. Related Works

## 6.1. High-dimensional Black-Box Optimization

Various approaches have been proposed to address high-dimensional black-box optimization. In Bayesian optimization (BO), some methods assume that objective function can be decomposed into a sum of several low-dimensional functions and train a large number of Gaussian processes (GPs) (Duvenaud et al., 2011; Kandasamy et al., 2015; Gardner et al., 2017). However, relying on training multiple

GPs reduces its scalability to a large number of evaluations. Other approaches assume that high-dimensional objective functions reside in a low-dimensional active subspace and introduce mapping to low-dimensional spaces (Chen et al., 2012; Garnett et al., 2014; Nayebi et al., 2019; Letham et al., 2020). However, these methods make strong assumptions that often fail to align with real-world problems.

Another line of BO methods utilizes local modeling or partitioning of the search space to address high dimensionality and scalability. TuRBO (Eriksson et al., 2019) fits multiple local models and restricts the search space to small trust regions to improve scalability. LA-MCTS (Wang et al., 2020) trains an unsupervised K-means classifier to partition the search space, identifies promising regions for sampling, and then employs BO-based optimizers such as BO or TuRBO. Including LA-MCTS, meta-algorithms that can be adapted to existing high-dimensional BO methods have also emerged. MCMC-BO (Yi et al., 2024) adopts Markov Chain Monte Carlo (MCMC) to adjust a set of candidate points towards more promising positions, and CMA-BO (Ngo et al., 2024) utilizes covariance matrix adaptation strategy to define a local region that has the highest probability of containing global optimum.

Evolutionary Algorithms (EAs) are also widely applied in high-dimensional black-box optimization. The most representative method, the covariance matrix adaptation evolution strategy (CMA-ES) (Hansen, 2006), operates by building and refining a probabilistic model of the search space. It iteratively updates the covariance matrix of a multivariate Gaussian distribution to capture the structure of promising solutions and uses feedback from the objective function to guide the search towards the optimal solution.

### 6.2. Generative Model-based Optimization

Several methods have been developed that utilize generative models to optimize black-box functions. Most approaches learn an inverse mapping from function values to the input domain and propose promising solutions by sampling from the trained model, conditioned on a high score (Brookes et al., 2019; Kumar & Levine, 2020; Krishnamoorthy et al., 2023; Wu et al., 2024; Kim et al., 2024b).

Building on the success of diffusion models, widely known for their expressivity in high-dimensional spaces (Ramesh et al., 2022; Ho et al., 2022), leveraging diffusion models for black-box optimization has also emerged (Krishnamoorthy et al., 2023; Wu et al., 2024; Kong et al., 2024; Yun et al., 2024). DDOM (Krishnamoorthy et al., 2023) trains a conditional diffusion model with classifier-free guidance (Ho & Salimans, 2021) and incorporates reweighted training to enhance the performance. Diff-BBO (Wu et al., 2024) trains an ensemble of conditional diffusion models, then employs an uncertainty-based acquisition function to select

the conditioning target value during candidate sampling.

Diff-BBO is closely related to our work, particularly in its use of diffusion models and its focus on online scenarios. However, utilizing multiple diffusion models results in a significant computational burden throughout the optimization process. Additionally, estimating uncertainty using diffusion models in high-dimensional spaces remains challenging, which can impact the effectiveness of uncertainty-based acquisition and may lead to suboptimal results. In contrast, our approach alleviates the computational burden by introducing a moving dataset and effectively scaling up to high-dimensional tasks with our posterior sampling strategy.

### 6.3. Amortized Inference in Diffusion Models

As diffusion models generate samples through a chain of stochastic transformations, sampling from the posterior distribution $p^{\text{post}}(\mathbf{x}) \propto p_\theta(\mathbf{x})r(\mathbf{x})$ is intractable. One of the widely used methods is estimating the guidance term by training a classifier on noised data (Dhariwal & Nichol, 2021; Lu et al., 2023). However, such data is unavailable in most cases, and it is often hard to train such a classifier in high-dimensional settings. Although Reinforcement learning (RL) methods have recently been proposed and shown interesting results (Black et al., 2024; Fan et al., 2024), naive RL fine-tuning does not provide an unbiased sampler of the target distribution (Uehara et al., 2024; Domingo-Enrich et al., 2024). To this end, we choose a relative trajectory balance proposed by Venkatraman et al. (2024) to obtain an unbiased sampler of the posterior distribution without training an additional classifier. Furthermore, we propose two post-processing strategies, local search and filtering, to improve the sample efficiency of our method.

## 7. Conclusion

In this work, we introduce DiBO, a novel generative model-based framework for high-dimensional and scalable black-box optimization. We repeat the process of training models and sampling candidates with diffusion models to find a global optimum in a sample-efficient manner. Specifically, by sampling candidates from the posterior distribution, we can effectively balance exploration and exploitation in high-dimensional spaces. We observe that our method surpasses various black-box optimization methods across synthetic and real-world tasks.

**Limitation and Future work.** While our method shows superior performance on a variety of tasks, we need to train the diffusion model with the updated dataset in every round. Although we analyze that our method exhibits low computational complexity compared to the state-of-the-art baselines, one may consider constructing a framework that can efficiently reuse the trained models from the previous rounds.

## Acknowledgement

This work is supported by the National Research Foundation of Korea (NRF) grant funded by the Korea government(MSIT) (No. RS-2025-00563763).

## Impact Statement

Real-world design optimization can lead to impressive breakthroughs, but it also carries certain risks. For example, while advanced pharmaceutical research could treat diseases once seen as incurable, those same methods could be misused to create harmful biochemical substances. Researchers have a responsibility to prioritize the well-being of society.

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

# A. Task Details

In this section, we present a detailed description of the benchmark tasks used in our experiments.

## A.1. Synthetic Functions

We conduct experiments on four complex synthetic functions that are widely used in BO literature: *Rastrigin, Ackley, Levy,* and *Rosenbrock*. Levy and Rosenbrock have global optima within long, flat valleys, whereas Rastrigin and Ackley have numerous local optima. This characteristic makes all four functions particularly challenging as the dimensionality increases. Following previous studies (Wang et al., 2020; Yi et al., 2024), we set the search space for each function as, Rastrigin: $[-5, 5]^D$, Ackley: $[-5, 10]^D$, Levy: $[-10, 10]^D$, and Rosenbrock: $[-5, 10]^D$.

## A.2. MuJoCo locomotion

MuJoCo locomotion task (Todorov et al., 2012) is a popular benchmark in Reinforcement Learning (RL). In this context, we optimize a linear policy $\mathbf{W}$ described by the equation $\mathbf{a} = \mathbf{W}\mathbf{s}$. The average return of this policy serves as our objective, and our goal is to identify the weight matrix that maximizes this return. We specifically focus on the *HalfCheetah* task, which has a dimensionality of 102. Each entry of the weight matrix $\mathbf{W}$ is constrained to the range $[-1, 1]$, and we utilize 3 rollouts for each evaluation. We followed the implementation of these tasks from the prior work Ngo et al. (2024). [2]

## A.3. Rover Trajectory Optimization

Rover Trajectory Optimization is a task determining the trajectory of a rover in a 2D environment suggested by Wang et al. (2018). Following previous work Ngo et al. (2024), we utilized a much harder version with a 100-dimensional variant, optimizing 50 distinct points. This task requires specifying a starting position $s$, a target position $g$, and a cost function applicable to the state space. We can calculate the cost $c(\mathbf{x})$ for a specific trajectory solution by integrating the cost function along the trajectory $\mathbf{x} \in [0, 1]^{100}$. The reward function is defined as: $f(\mathbf{x}) = c(\mathbf{x}) + \lambda (\|\mathbf{x}_{0,1} - s\|_1 + \|\mathbf{x}_{99,100} - g\|_1) + b$. We followed implementation from Wang et al. (2018). [3]

## A.4. LassoBench

LassoBench (Šehić et al., 2022) [4] is a challenge focused on optimizing the hyperparameters of Weighted LASSO (Least Absolute Shrinkage and Selection Operator) regression. The goal is to fine-tune a set of hyperparameters to achieve a balance between least-squares estimation and the sparsity-inducing penalty term. LassoBench serves both synthetic (simple, medium, high, hard) and real-world tasks, including (Breast cancer, Diabetes, Leukemia, DNA, and RCV1). Specifically, we focused on the DNA task, which is a 180-dimensional hyperparameter optimization task that utilizes a DNA dataset from a microbiological study. In Figure 4, we present the original results multiplied by -1 for improved visibility.

---

[2] https://github.com/LamNgo1/cma-meta-algorithm
[3] https://github.com/zi-w/Ensemble-Bayesian-Optimization
[4] https://github.com/ksehic/LassoBench

# B. Methodology Details

In this section, we provide a detailed overview of the methodology, covering model implementations and architectures, training procedures, hyperparameter settings, and computational resources.

## B.1. Training Models

### B.1.1. TRAINING PROXY MODEL

We train five ensembles of proxies. To implement the proxy function, we use MLP with three hidden layers, each consisting of 256 (512 for 400 dim tasks) hidden units and GELU (Hendrycks & Gimpel, 2016) activations. We train a proxy model using Adam (Kingma, 2015) optimizer for 50 (100 for 400 dim tasks) epochs per round, with a learning rate $1 \times 10^{-3}$. We set the batch size to 256. The hyperparameters related to the proxy are listed in Table 1.

*Table 1.* Hyperparameters for Training Proxy

|              | Parameters         | Values                      |
| ------------ | ------------------ | --------------------------- |
| Architecture | Num Ensembles      | 5                           |
|              | Number of Layers   | 3                           |
|              | Num Units          | 256 (Default) / 512 (400D)  |
| Training     | Batch size         | 256                         |
|              | Optimizer          | Adam                        |
|              | Learning Rate      | $1 \times 10^{-3}$          |
|              | Training Epochs    | 50 (Default) / 100 (400D)   |

### B.1.2. TRAINING DIFFUSION MODEL

We utilize the temporal Residual MLP architecture from Venkatraman et al. (2024) as the backbone of our diffusion model. The architecture consists of three hidden layers, each containing 512 hidden units. We implement GELU activations alongside layer normalization (Ba, 2016). During training, we use the Adam optimizer for 50 epochs (100 for 400 dim tasks) per round with a learning rate of $1 \times 10^{-3}$. We set the batch size to 256. We employ linear variance scheduling and noise prediction networks with 30 diffusion steps for all tasks. The hyperparameters related to the diffusion model are summarized in Table 2.

*Table 2.* Hyperparameters for Training Diffusion Models

|                    | Parameters        | Values                     |
| ------------------ | ----------------- | -------------------------- |
| Architecture       | Number of Layers  | 3                          |
|                    | Num Units         | 512                        |
| Training           | Batch size        | 256                        |
|                    | Optimizer         | Adam                       |
|                    | Learning Rate     | $1 \times 10^{-3}$         |
|                    | Training Epochs   | 50 (Default) / 100 (400D)  |
| Diffusion Settings | Num Timesteps     | 30                         |

## B.2. Sampling Candidates

### B.2.1. FINE-TUNING DIFFUSION MODEL

We use relative trajectory balance (RTB) loss to fine-tune the diffusion model for obtaining an amortized sampler of the posterior distribution.

$$\mathcal{L}(\mathbf{x}_{0:T}; \psi) = \left( \log \frac{Z_\psi \cdot p_\psi(\mathbf{x}_{0:T})}{\exp\left(\beta \cdot r_\phi(\mathbf{x}_0)\right) \cdot p_\theta(\mathbf{x}_{0:T})} \right)^2 \tag{17}$$

As stated in the original work by Venkatraman et al. (2024), the gradient of this objective concerning $\psi$ does not necessitate backpropagation into the sampling process that generates a trajectory $\mathbf{x}_{0:T}$. Consequently, the loss can be optimized in an off-policy manner. Specifically, we can optimize Equation (17) with (1): on-policy trajectories $\mathbf{x}_{0:T} \sim p_\psi(\mathbf{x}_{0:T})$ or (2): off-policy trajectories $\mathbf{x}_{0:T}$ generated by noising process given $\mathbf{x}_0$ sampled from the buffer.

To effectively fine-tune our diffusion model, we train $p_\psi$ using both methods. For each iteration, we select a batch of on-policy trajectories with a probability of 0.5 and off-policy trajectories otherwise. When sampling from the buffer, we use reward-prioritized sampling to focus on data points with high UCB scores. We conducted additional analysis on off-policy training in Appendix D.3.

We initialize $\psi \leftarrow \theta$ with each iteration, so the architecture and diffusion timestep is the same with Table 2. During training, we use Adam optimizer for 50 epochs (100 epochs for 400D) with learning rate $1 \times 10^{-4}$. We set the batch size to 256. The hyperparameters for fine-tuning the diffusion model are summarized in Table 3.

*Table 3.* Hyperparameters for Finetuning Diffusion Models

|  | Parameters | Values |
|---|---|---|
| Architecture | Number of Layers | 3 |
|  | Num Units | 512 |
| Training | Batch size | 256 |
|  | Optimizer | Adam |
|  | Learning Rate | $1 \times 10^{-4}$ |
|  | Training Epochs | 50 (Default) / 100 (400D) |
| Diffusion Settings | Num Timesteps | 30 |

All the training is done with a Single NVIDIA RTX 3090 GPU.

### B.2.2. ESTIMATING MARGINAL LIKELIHOOD

During the **local search** and **filtering** in Section 3.2, we use the probability flow ordinary differential equation (PF ODE) to estimate the marginal log-likelihood of the diffusion prior $\log p_\theta(\mathbf{x})$. We consider the diffusion forward process as the following stochastic differential equation (SDE):

$$d\mathbf{x} = \mathbf{f}(\mathbf{x}, t)dt + g(t)d\mathbf{w} \tag{18}$$

and the corresponding reverse process is

$$d\mathbf{x} = [\mathbf{f}(\mathbf{x}, t) - g(t)^2 \nabla_\mathbf{x} \log p_t(\mathbf{x})]dt + g(t)d\bar{\mathbf{w}} \tag{19}$$

where $\mathbf{w}$ and $\bar{\mathbf{w}}$ are forward and reverse Brownian motions, and $\mathbf{f}$ and $g$ are drift coefficient and diffusion coefficient respectively. The quantity $p_t(\mathbf{x})$ denotes the marginal distribution of $\mathbf{x}$ at time $t$. As we do not have direct access to score $\nabla_\mathbf{x} \log p_t(\mathbf{x})$, it should be modeled with network approximation $s_\theta(\mathbf{x}, t) \approx \nabla_\mathbf{x} \log p_t(\mathbf{x})$, while in our case implicitly modeled by noise prediction network $\epsilon_\theta(\mathbf{x}, t)$ (Kingma et al., 2021).

There also exists a deterministic PF ODE,

$$d\mathbf{x} = [\mathbf{f}(\mathbf{x}, t) - \frac{1}{2}g(t)^2 \nabla_\mathbf{x} \log p_t(\mathbf{x})]dt \tag{20}$$

which evolves the sample $\mathbf{x}$ through the same marginal distributions $\{p_t(\mathbf{x})\}$ as Equations (18) and (19), under suitable regularity conditions. (Song et al., 2021)

With the trained $s_\theta(\mathbf{x}, t)$, we can estimate $\log p_0(\mathbf{x}_0) = \log p_\theta(\mathbf{x})$ by applying the instantaneous change-of-variables formula (Chen et al., 2018) to the PF ODE:

$$\log p_0(\mathbf{x}_0) = \log p_T(\mathbf{x}_T) + \int_0^T \nabla \cdot \bar{\mathbf{f}}_\theta(\mathbf{x}(t), t)\, dt \tag{21}$$

where

$$\bar{\mathbf{f}}_\theta(\mathbf{x}(t), t) := \mathbf{f}(\mathbf{x}, t) - \frac{1}{2}g(t)^2 s_\theta(\mathbf{x}, t). \tag{22}$$

However, directly computing the trace of $\bar{\mathbf{f}}_\theta$ is computationally expensive. Following Grathwohl et al. (2019); Song et al. (2021), we use the Skilling-Hutchinson trace estimator (Skilling, 1989; Hutchinson, 1989) to estimate the trace efficiently:

$$\nabla \cdot \bar{\mathbf{f}}_\theta(\mathbf{x}, t) = \mathbb{E}_\nu[\nu^\mathsf{T} \nabla \bar{\mathbf{f}}_\theta(\mathbf{x}, t)\nu], \tag{23}$$

where the $\nu$ is sampled from the Rademacher distribution.

We solve Equation (21) using a differentiable ODE solver `torchdiffeq` (Chen, 2018) with 4th-order Runge–Kutta (RK4) integrator, accumulating the divergence term in the integral to approximate $\log p_0(\mathbf{x}_0)$. Since the PF ODE is deterministic, this entire simulation is fully differentiable, enabling gradient-based optimization with respect to the $\mathbf{x}_0$, thereby supporting the local search stage.

### B.2.3. HYPERPARAMETERS

For the upper confidence bound (UCB), we fixed $\gamma = 1.0$ that controls the exploration-exploitation. For the target posterior distribution, the inverse temperature parameter $\beta$ controls the trade-off between the influence of $\exp(r_\phi(\mathbf{x}))$ and $p_\theta(\mathbf{x})$. When selecting querying candidates, we sample $M = B \times 10^2$ candidates from $p_\psi(\mathbf{x})$, perform a local search for $J$ steps, and retain $B$ candidates for batched querying. After querying and adding candidates, we maintain our training dataset to contain $L$ high-scoring samples. We present the detailed hyperparameter settings in Table 4. We also conduct several ablation studies to explore the effect of each hyperparameter on the performance.

*Table 4.* Hyperparameters during sampling candidates

|  | Inverse Temperature $\beta$ | Local Search Steps $J$ | Buffer Size $L$ |
|---|---|---|---|
| Synthetic 200D | $10^5$ | 10 | 1000 (Rastrigin) / 500 (Others) |
| Synthetic 400D | $10^5$ | 15 | 1000 (Rastrigin) / 500 (Others) |
| HalfCheetah 102D | $10^4$ | 10 | 300 |
| RoverPlanning 100D | $10^5$ | 30 | 300 |
| DNA 180D | $10^5$ | 50 | 300 |

# C. Baseline Details

In this section, we provide details of the baseline implementation and hyperparameters used in our experiments.

**TuRBO** (Eriksson et al., 2019): We use the original code [5] and keep all settings identical to those in the original paper. For all the algorithms utilizing TuRBO as a base algorithm (TuRBO, LA-MCTS, MCMC-BO), we use TuRBO-1 (No parallel local models).

**LA-MCTS** (Wang et al., 2020): We use the original code [6] and keep all settings identical to those in the original paper.

**MCMC-BO** (Yi et al., 2024): We utilize the original code [7] and adjust the standard deviation of the proposal distribution during the Metropolis-Hastings steps to replicate the results from the original paper. Due to memory constraints during the MCMC steps, we report a total of 6,000 evaluations for the $D = 400$ tasks.

**CMA-BO** (Ngo et al., 2024): We use the original code [8] and keep all settings identical to those in the original paper.

**CbAS** (Brookes et al., 2019): We reimplement the original code [9] with PyTorch and keep all settings identical to those in the original paper.

**MINs** (Kumar & Levine, 2020): We reimplement the code from Trabucco et al. (2022) [10] with PyTorch and keep all settings identical to those in the original paper.

**DDOM** (Krishnamoorthy et al., 2023): To ensure a fair comparison, we reimplement the original code [11] to work with our diffusion models and add classifier free guidance for conditional generation. We use the same method-specific hyperparameters following the original paper and tune the training epochs (200 as a default, 400 for $D = 400$ tasks) for each task to optimize performance.

**Diff-BBO** (Wu et al., 2024): As there is no open-source code, we implement it according to the details provided in the original paper. As with DDOM, we use the original method-specific hyperparameters and tune the training epochs (200 as a default, 400 for $D = 400$ tasks) for each task to improve performance.

**CMA-ES** (Hansen, 2006): We use an existing library pycma (Hansen et al., 2019) and adjust the initial standard deviation as $\sigma_0 = 0.1$, which gives better performance on all tasks.

---

[5] https://github.com/uber-research/TuRBO
[6] https://github.com/facebookresearch/LA-MCTS
[7] https://drive.google.com/drive/folders/1fLUHIduB3-pR78Y1YOhhNtsDegaOqLNU?usp=sharing
[8] https://github.com/LamNgo1/cma-meta-algorithm
[9] https://github.com/dhbrookes/CbAS
[10] https://github.com/brandontrabucco/design-bench
[11] https://github.com/siddarthk97/ddom

# D. Extended Additional Analysis

In this section, we present additional analysis on DiBO that is not included in the main manuscript due to the page limit.

## D.1. Effect of Training Epochs

The number of epochs for training models can be crucial in the performance of black-box optimization algorithms. If we use too small a number of epochs, the proxy may underfit, and the diffusion model may find it hard to capture the complex data distribution accurately. On the other hand, if we use too large a number of epochs, the proxy and the diffusion may overfit to the dataset, and the overall procedure takes longer time for each round.

To this end, we conduct experiments on Rastrigin-200D and HalfCheetah-102D by varying training epochs. As shown in the Figure 6, when we use large training epochs, the performance improves significantly at the early stage but eventually converges to the sub-optimal results due to the overfitting of the proxy and diffusion model, which may hinder exploration towards promising regions.

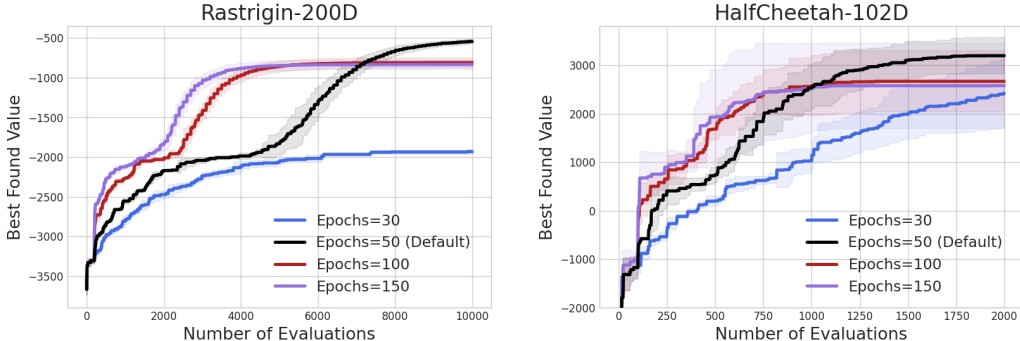

*Figure 6.* Performance of DiBO in Rastrigin-200D and HalfCheetah-102D by varying training epochs. Experiments are conducted with four random seeds. Mean and one standard deviation are reported.

## D.2. Analysis on Local Search Steps $J$

We conduct additional analysis on local search steps $J$. Through local search, we can capture the modes of the target distribution, which leads to high sample efficiency. However, using too large local search steps may focus on exploiting a single mode with the highest density of the target distribution, resulting in sub-optimal results.

To this end, we conduct experiments on Rastrigin-200D and HalfCheetah-102D by varying $J$. As shown in the Figure 7, our method shows a relatively slow learning curve when we remove the local search. On the other hand, if we use too large $J$, it struggles to escape from local optima and eventually results in sub-optimal results.

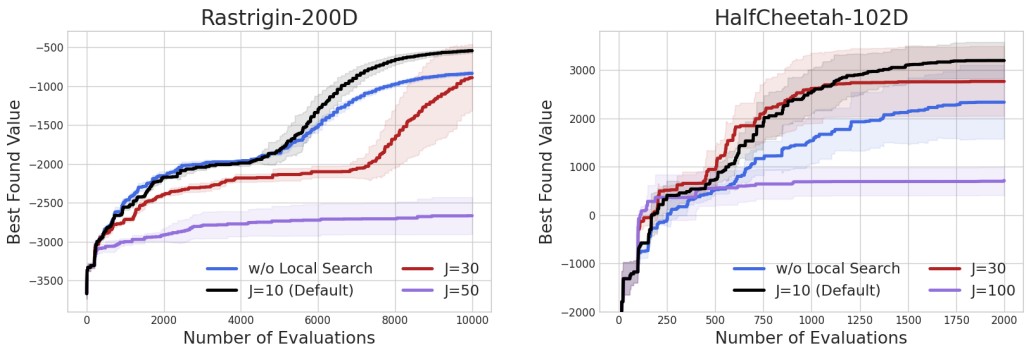

*Figure 7.* Performance of DiBO in Rastrigin-200D and HalfCheetah-102D by varying $J$. Experiments are conducted with four random seeds. Mean and one standard deviation are reported.

## D.3. Effect of Off-policy Training in Amortized Inference

During the fine-tuning stage, we employ off-policy training with the RTB loss function, as detailed in Appendix B.2.1. To assess the impact of this approach, we conduct a comparative experiment using only on-policy training.

As illustrated in Figure 8, off-policy training demonstrates a significant performance advantage over on-policy training. In on-policy training, the model is restricted to learning only from the generated samples. Consequently, crucial data points, particularly those associated with significant events, are rarely encountered during training. In contrast, off-policy training effectively captures these critical regions by directly leveraging information from a replay buffer, enabling the model to learn from informative data points efficiently.

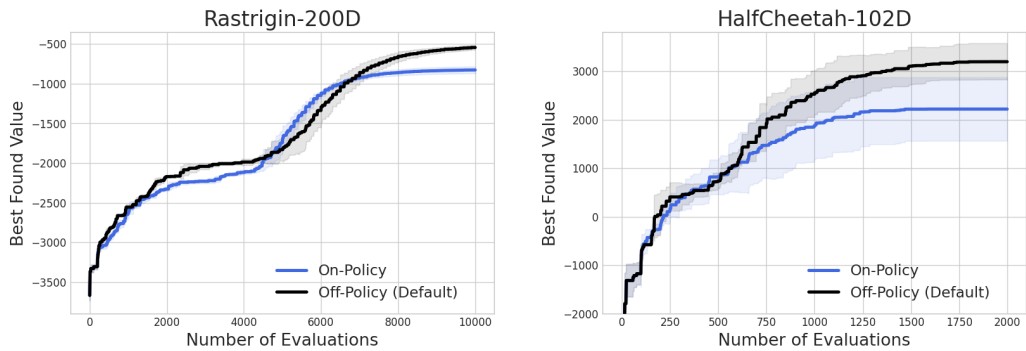

*Figure 8.* Performance of DiBO in Rastrigin-200D and HalfCheetah-102D with and without off-policy training. Experiments are conducted with four random seeds. Mean and one standard deviation are reported.

## D.4. Analysis on Initial Dataset Size $|\mathcal{D}_0|$

The size of the initial dataset $|\mathcal{D}_0|$ can be crucial in the performance of black-box optimization algorithms. If the initial dataset is too small and concentrates on a small region compared to the whole search space, it is hard to explore diverse promising regions without proper exploration strategies.

To this end, we conduct experiments by varying $|\mathcal{D}_0|$ on the synthetic tasks. As shown in the Figure 9, our method demonstrates robustness regarding the size of the initial dataset. It indicates that our exploration strategy, proposing candidates by sampling from the posterior distribution, is powerful for solving practical high-dimensional black-box optimization problems.

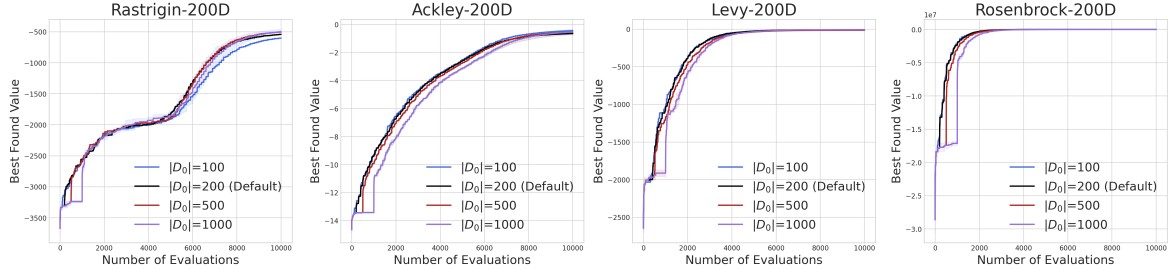

*Figure 9.* Performance of DiBO in synthetic functions by varying $|\mathcal{D}_0|$. Experiments are conducted with four random seeds. Mean and one standard deviation are reported.

## D.5. Analysis on Batch Size $B$

The batch size $B$ can be crucial in the performance of black-box optimization algorithms. As the number of evaluations is mostly limited, if we use too large $B$, it is hard to focus on high-scoring regions. On the other hand, if we use too small $B$, it hinders exploration, and it is hard to escape from local optima.

To this end, we conduct experiments by varying $B$ on the synthetic tasks. Note that we fix the batch size for all main experiments as $B = 100$. We visualize the experiment results in Figure 10. We can observe that our method shows robust performance across different $B$ while using a large batch size leads to slightly slow convergence compared to using a small batch size. Using a smaller batch size shows better sample efficiency but also leads to an increase in computational time.

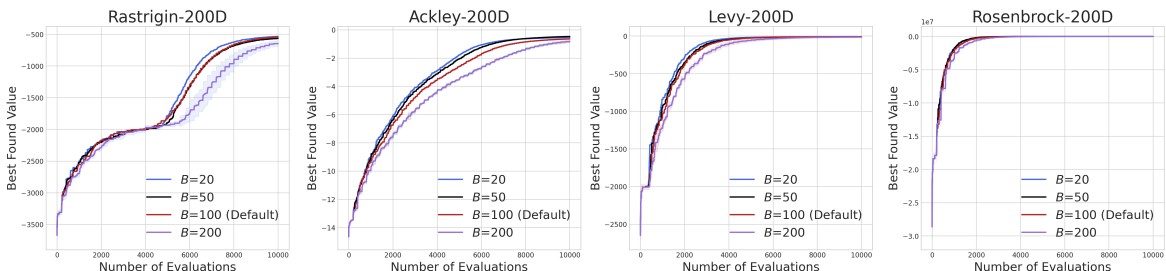

*Figure 10.* Performance of DiBO in synthetic functions by varying $B$. Experiments are conducted with four random seeds. Mean and one standard deviation are reported.

### D.6. Analysis on Uncertainty Estimation

To promote exploration, we estimate uncertainty with an ensemble of proxies and adopt an upper confidence bound (UCB) to define the target distribution. Specifically, we use: $r_\phi(\mathbf{x}) = \mu_\phi(\mathbf{x}) + \gamma \cdot \sigma_\phi(\mathbf{x})$, where $\gamma$ controls the degree of uncertainty bonus. We evaluated two aspects of this approach in the HalfCheetah-102D task.

To analyze the effectiveness of the ensemble strategy for uncertainty estimation, besides our ensemble method, we test Monte Carlo (MC) dropout (Gal & Ghahramani, 2016) and a setup without uncertainty estimation (one proxy). As shown in Figure 11a, the ensemble strategy effectively estimates the uncertainty and improves sample efficiency compared to others.

To analyze if UCB with uncertainty bonus promotes exploration, we conduct experiments by varying the parameter $\gamma$ and analyzing its impact on performance. Figure 11b demonstrate that increasing $\gamma$ leads to more extensive search space exploration. However, excessively large $\gamma$ values ($\gamma = 10.0$) dilute the focus on exploitation, slowing convergence.

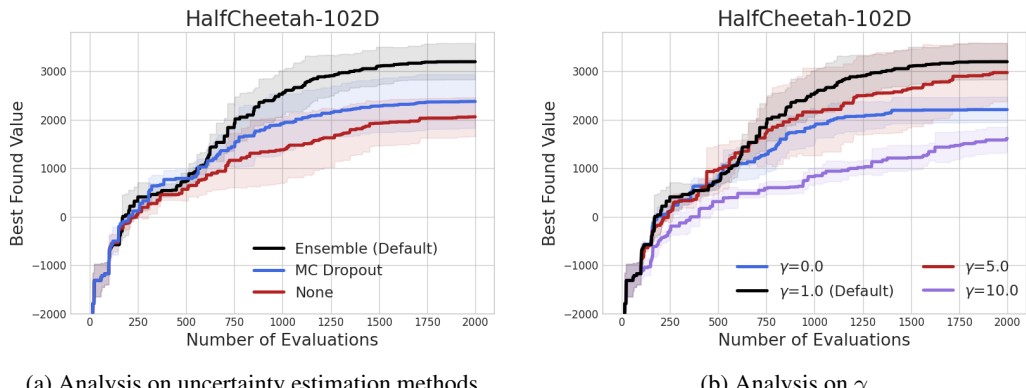

(a) Analysis on uncertainty estimation methods     (b) Analysis on $\gamma$

*Figure 11.* Performance of DiBO in HalfCheetah-102D with varying uncertainty estimation methods and gamma $\gamma$. Experiments are conducted with four random seeds. Mean and one standard deviation are reported.

These findings demonstrate that an ensemble of proxies effectively captures uncertainty even in high-dimensional spaces. Moreover, designing a target distribution that incorporates UCB helps balance exploration and exploitation, improving sample efficiency during optimization.

### D.7. Time Complexity of our method

We report the average running time per each round in Table 5. All training is done with a single NVIDIA RTX 3090 GPU and Intel Xeon Platinum CPU @ 2.90GHZ. As shown in the table, the running time of our method is similar to generative model-based approaches and mostly faster than BO methods. It demonstrates the efficacy of our proposed method.

*Table 5.* Average time (in seconds) for each round in each method.

|  | Rastrigin-200D | Rastrigin-400D | Ackley-200D | Ackley-400D | Levy-200D | Levy-400D | Rosenbrock-200D | Rosenbrock-400D |
|---|---|---|---|---|---|---|---|---|
| TuRBO | 244.39 ± 0.21 | 1089.09 ± 0.04 | 33.91 ± 0.24 | 41.50 ± 0.06 | 167.58 ± 0.21 | 59.70 ± 0.02 | 452.21 ± 0.15 | 45.05 ± 0.01 |
| LA-MCTS | 222.95 ± 6.59 | 256.82 ± 8.89 | 150.27 ± 3.97 | 184.14 ± 1.67 | 90.47 ± 1.69 | 229.80 ± 11.99 | 154.56 ± 4.08 | 223.59 ± 7.52 |
| MCMC-BO | 341.98 ± 3.54 | 429.02 ± 4.61 | 370.03 ± 4.17 | 345.60 ± 3.42 | 337.87 ± 3.65 | 429.02 ± 4.61 | 419.07 ± 5.12 | 448.56 ± 5.13 |
| CMA-BO | 643.14 ± 4.01 | 833.97 ± 2.12 | 661.15 ± 4.52 | 854.23 ± 2.77 | 694.67 ± 4.59 | 871.33 ± 3.66 | 645.65 ± 4.95 | 857.39 ± 2.53 |
| CbAS | 212.69 ± 5.27 | 213.64 ± 5.79 | 207.35 ± 4.01 | 213.67 ± 6.62 | 212.61 ± 4.05 | 212.69 ± 9.83 | 203.87 ± 1.43 | 221.68 ± 5.01 |
| MINs | 28.36 ± 0.45 | 32.20 ± 0.12 | 28.83 ± 0.41 | 29.14 ± 0.60 | 29.91 ± 0.17 | 29.95 ± 0.37 | 30.22 ± 1.02 | 28.81 ± 0.67 |
| DDOM | 23.74 ± 0.28 | 26.57 ± 0.11 | 23.53 ± 0.04 | 26.62 ± 0.16 | 23.40 ± 0.15 | 26.46 ± 0.15 | 23.19 ± 0.12 | 26.55 ± 0.15 |
| Diff-BBO | 128.75 ± 0.89 | 143.80 ± 1.16 | 131.16 ± 0.86 | 143.96 ± 1.68 | 130.07 ± 0.74 | 143.97 ± 1.79 | 128.33 ± 1.74 | 144.03 ± 1.58 |
| CMA-ES | 0.03 ± 0.01 | 0.05 ± 0.00 | 0.03 ± 0.00 | 0.04 ± 0.00 | 0.04 ± 0.00 | 0.05 ± 0.00 | 0.03 ± 0.00 | 0.04 ± 0.00 |
| **DiBO** | 42.74 ± 0.26 | 79.63 ± 1.07 | 39.72 ± 0.18 | 71.99 ± 0.10 | 39.55 ± 0.10 | 72.21 ± 0.53 | 39.57 ± 0.28 | 72.88 ± 0.34 |

### D.8. Analysis on more computation time

In this section, we present supplementary results demonstrating that more computing time can significantly improve performance beyond the results reported in the main text. For example, on the Ackley-200D benchmark, with the local search steps $J = 50$ and buffer size, $L = 2000$ improves performance from the default configuration value of $-0.643$ to a score of $-0.260$. While increasing the number of local search steps has a possibility of converging to the local optimum, a large buffer size complements this issue. However, using larger $J$ and $L$ leads to an increase in computational complexity. Exploring methods that reduce the complexity of training while maintaining large buffer sizes and local search steps may be a promising research direction. We leave it as a future work.

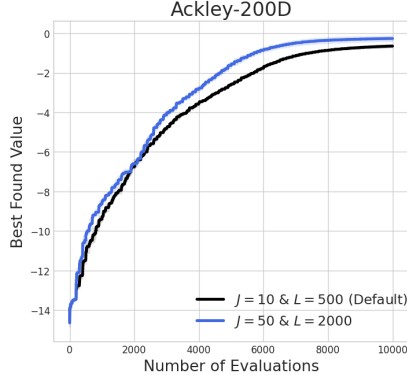

*Figure 12.* Performance of DiBO in Ackley-200D with local search epochs $J = 50$, and buffer size $L = 2000$. Experiments are conducted with four random seeds. Mean and one standard deviation are reported.

