# OpenReview forum: "Posterior Inference with Diffusion Models for High-dimensional Black-box Optimization"
_ICML.cc/2025/Conference — ICML 2025 poster_

### Official Review · Reviewer_DZqC · 2025-03-10

**Overall Recommendation:** 3

**Summary:**

## Summary
* The authors propose a two-stage approach for black box optimization using diffusion model. The first stage is training stage. The authors propose to train a weighted unconditional model for density estimation, and an ensemble of proxy models to capture the value and uncertainty of target. This diffusion model + discriminative approach is commonly adopted, e.g. in classifier based guidance.
* Next, the authors propose to findtune the model (amortized inference) using a relative trajectory balance. The target is designed to balance exploration and exploitation. To further enhance the performance, the authors adopts two post processing technique: local search and filtering. The local search is a gradient ascent on finetuning target, and the filtering is a selection of candidates.
* Empirical results on multiple dataset show the effectiveness of their approach.

**Claims And Evidence:**

## Claims And Evidence
* The reweighting scheme is interesting. However, the score matching loss of diffusion model is not a direct max likelihood target (See [Maximum Likelihood Training of Score-Based Diffusion Models]) Therefore, whether simple reweighted training achieve weighted likelihood such as Eq. 11 remains questionable.

**Essential References Not Discussed:**

## Essential References Not Discussed
* The reference to prior works is sufficient.

**Experimental Designs Or Analyses:**

## Experimental Designs Or Analyses
* The experimental result and analysis are sufficient. Abundant results show the effectiveness of the proposed method, in terms of performance and complexity. The effectiveness of different component proposed by the authors is also verified.

**Methods And Evaluation Criteria:**

## Methods And Evaluation Criteria
* The benchmark datasets look standard for this field and the ablation studys are sufficient to support the claims.

**Other Comments Or Suggestions:**

-

**Other Strengths And Weaknesses:**

* One part that I like about this paper is its empirical evaluation. Through benchmark and real world problems, the authors successfully show the advantage of their approach over prior works.
* One part that I do not like about this paper is that it contains to many sub-parts and tricks. The paper is driven by a clear performance target, but there is no clear technical clue that lead this paper.

**Questions For Authors:**

-

**Relation To Broader Scientific Literature:**

## Relation To Broader Scientific Literature
* The proposed approach will likely to be a strong baseline in black box optimization.

**Theoretical Claims:**

## Theoretical Claims
* There are no theoretical claims.

---

> ### Author Rebuttal · Authors · 2025-03-31
>
> Thank you for your positive assessment of our paper's extensive experiment results. We've attempted to answer your questions below.
>
> >**Claims And Evidence)** Therefore, whether simple reweighted training achieve weighted likelihood such as Eq. 11 remains questionable.
>
> Thank you for pointing out the question regarding Eq.11. As you mentioned, we try to maximize the ELBO instead of the marginal likelihood. We will fix the Eq.11 in the final manuscript.
>
> > **Other Strengths And Weaknesses)** One part that I do not like about this paper is that it contains to many sub-parts and tricks. The paper is driven by a clear performance target, but there is no clear technical clue that lead this paper.
>
> Thank you for your constructive feedback. While there are several sub-parts in our method, please note that we systemically analyze the effect of each component through extensive ablation studies to verify that each component is crucial for improving performance.
>
> Furthermore, please note that several ideas we imported in this paper are already considered as a reasonable choice for training diffusion models as an amortized sampler effectively. For example, off-policy training is suggested in various GFlowNets literature [1, 2, 3].
>
> [1] Venkatraman, Siddarth, et al. "Amortizing intractable inference in diffusion models for vision, language, and control."
>
> [2] Akhound-Sadegh, Tara, et al. "Iterated denoising energy matching for sampling from boltzmann densities."
>
> [3] Rector-Brooks, Jarrid, et al. "Steering masked discrete diffusion models via discrete denoising posterior prediction."
>
> Thank you again for your comments. We hope we have addressed them satisfactorily above, but do not hesitate to let us know if you have further questions. We are always ready to engage in further discussion!

---

### Official Review · Reviewer_jXRP · 2025-03-13

**Overall Recommendation:** 4

**Summary:**

This paper proposes a novel high-dimensional black-box optimization method, where the authors train a diffusion model based on the weighted data as the prior and performs posterior sampling when combined with a uncertainty-aware function proxy. The authors also use local search and filtering strategies to further refine the posterior samples. Over extensive benchmarks the proposed DiBO demonstrates improved optimization performances compared to representative high-dimensional optimization methods.

**Claims And Evidence:**

The main contributions that the authors claim:

1. The proposed diffusion-based algorithm to address the scalability and efficiency in high-dimensional optimization.

2. Superior performances over a variety of tasks compared to state-of-the art baselines.

I think the proposed method and the experimental results support the claimed contributions.

**Essential References Not Discussed:**

I think essential references are discussed.

**Experimental Designs Or Analyses:**

I think the experiment setting and ablation studies are comprehensive.

**Methods And Evaluation Criteria:**

I think the method makes sense and the used benchmarks are representative in high-dimensional black-box optimization.

**Other Comments Or Suggestions:**

I think it is a good work which well applies diffusion models for high-dimensional black-box optimization.

Given the expressive power of diffusion model, I think some tasks including structured input space may further enhance the paper (e.g. chemical/protein design).

**Other Strengths And Weaknesses:**

The paper is clear and well-written. And I don't have major concerns in terms of the weakness.

**Questions For Authors:**

How to set the number of ensembles to guarantee the uncertainty quantification is reasonable?

**Relation To Broader Scientific Literature:**

I think the idea of incorporating diffusion model for input prior learning and casting the sampling as posterior inference is novel and a suitable usage of diffusion model to address high-dimensional issues.

**Theoretical Claims:**

Not applicable.

---

> ### Author Rebuttal · Authors · 2025-03-31
>
> Thank you for your positive comment and for considering our key idea, incorporating the diffusion model as prior and casting sampling as posterior inference for solving high-dimensional black-box optimization, as novel. We answer your questions below.
>
> >**Other Comments Or Suggestions)** Given the expressive power of diffusion model, I think some tasks including structured input space may further enhance the paper (e.g. chemical/protein design).
>
> As you mentioned, tasks including structured input space further enhance the usefulness of our method. To this end, we conduct experiments on molecular optimization following [1]. As shown in the table, our method achieves not only higher performance but also high sample efficiency compared to recent BO baselines for structured inputs. We promise to add these results in our final manuscript.
>
>
> **Experiment results on structured inputs. Experiments are conducted with four random seeds.**
> | Tasks|# Evaluation Budget |LOL-BO [1]|CoBO [2]|DiBO|
> |:-|-|:-|:-|:-:|
> |Zaleplon MPO|20000|0.711 ± 0.014|0.724 ± 0.004|0.739 ± 0.034|
> ||30000|0.723 ± 0.006|0.728 ± 0.002|0.771 ± 0.002|
> ||40000|0.739 ± 0.000|0.738 ± 0.002|0.771 ± 0.002|
> |Perindopril MPO | 20000|0.734 ± 0.000|0.715 ± 0.025|0.815 ± 0.004|
> ||30000|0.771 ± 0.014|0.788 ± 0.024|0.818 ± 0.006|
> ||40000|0.798 ± 0.021|0.796 ± 0.018|0.825 ± 0.009|
>
> [1] Maus, Natalie, et al. "Local latent space bayesian optimization over structured inputs."
>
> [2] Lee, Seunghun, et al. "Advancing bayesian optimization via learning correlated latent space."
>
> > **Questions For Authors)** How to set the number of ensembles to guarantee the uncertainty quantification is reasonable?
>
> Thank you for your interest in our work. The number of ensembles is crucial to reasonably quantify the uncertainty of the surrogate model. To this end, we conduct experiments by varying number of ensembles, $K$. As shown in the table, there is no big difference in performance when we increase $K$ more than $K=5$. However, without uncertainty quantification or too small number of ensembles leads to poor performance, which indicates that uncertainty quantification is crucial for high-dimensional black-box optimization problems.
>
> **Ablation studies on the number of ensembles ($K$). Experiments are conducted with four random seeds.**
>
> |             | $K$         | DiBO (Ours) |
> |:----------- |:----------- |:----------- |
> | HalfCheetah | 1 (None)    | 2750.765    |
> |             | 3           | 2604.994    |
> |             | 5 (Default) | 3191.215    |
> |             | 7           | 3131.849    |
> |             | 9           | 2926.619    |

---

### Official Review · Reviewer_rUT1 · 2025-03-17

**Overall Recommendation:** 2

**Summary:**

This paper utilizes the diffusion model for high-dimensional black-box optimization. At each iteration, they sample the candidates from the posterior distribution. The empirical results show that the proposed method outperforms other baselines.

**Claims And Evidence:**

The authors claim that by sampling candidates from the posterior distribution, the proposed method can effectively balance exploration and exploitation. However, they only measure the uncertainty of a portion of their model, which is from the ensemble of proxies. There is no measurement regarding the uncertainty of the generative model or discussion about the relationships between two terms of uncertainty. It remains unclear why this sampling approach can effectively balance exploration and exploitation. And there is no theoretical guarantee provided.

**Essential References Not Discussed:**

Please refer to the previous sections to add the references.

**Experimental Designs Or Analyses:**

My concern is that many baselines for high-dimensional black-box optimization are missing. I conducted a brief literature search and listed some of them [1-9]. DDOM is not an appropriate baseline as it is designed for offline optimization. For the ablation study in the appendix, it will be helpful if the authors can include other baselines in the analysis of batch size and initial dataset size. I assume the performance of DiBO will degrade when there is insufficient data for the diffusion model to learn or update the data distribution.

[1] Ament, Sebastian, et al. "Unexpected improvements to expected improvement for bayesian optimization."
[2] Eriksson, David, and Martin Jankowiak. "High-dimensional Bayesian optimization with sparse axis-aligned subspaces."
[3] Nayebi, Amin, Alexander Munteanu, and Matthias Poloczek. "A framework for Bayesian optimization in embedded subspaces."
[4] Wang, Zi, et al. "Batched large-scale Bayesian optimization in high-dimensional spaces."
[5] Wang, Linnan, Rodrigo Fonseca, and Yuandong Tian. "Learning search space partition for black-box optimization using monte carlo tree search."
[6] Letham, Ben, et al. "Re-examining linear embeddings for high-dimensional Bayesian optimization."
[7] Song, Lei, et al. "Monte carlo tree search based variable selection for high dimensional bayesian optimization."
[8] Ziomek, Juliusz Krzysztof, and Haitham Bou Ammar. "Are random decompositions all we need in high dimensional Bayesian optimisation?."
[9] Nguyen, Quan, et al. "Local Bayesian optimization via maximizing probability of descent."

**Methods And Evaluation Criteria:**

Both the synthetic and real-world benchmark datasets have been evaluated. The authors follow the standard problem setting of high-dimensional bayesian optimization. Since the authors claim their method can effectively capture complex and multi-modal data distribution. It is helpful to verify that by including the black-box optimization benchmarks with structured data in the design space, such as molecular optimization tasks [1].

[1] Maus, Natalie, et al. "Local latent space bayesian optimization over structured inputs."

**Other Comments Or Suggestions:**

Please refer to the previous sections.

**Other Strengths And Weaknesses:**

Please refer to the previous sections.

**Questions For Authors:**

Please refer to the previous sections.

**Relation To Broader Scientific Literature:**

This paper extends the diffusion model from offline to online black-box optimization, which has been studied in Diff-BBO [1]. The posterior sampling approaches of the proposed algorithm appears similar to the ones used in offline optimization with diffusion models [2,3].

[1] Wu, Dongxia, et al. "Diff-BBO: Diffusion-Based Inverse Modeling for Black-Box Optimization."
[2] Yu, Peiyu, et al. "Latent energy-based odyssey: Black-box optimization via expanded exploration in the energy-based latent space."
[3] Kong, Lingkai, et al. "Diffusion models as constrained samplers for optimization with unknown constraints."

**Theoretical Claims:**

There is no theoretical guarantee provided for the sampling approach. It would be helpful if the authors can include some theoretical analysis of their proposed algorithm. For example, can it be proven that the proposed approach guarantees an optimal or near-optimal solution for black-box optimization under certain assumptions?

---

> ### Author Rebuttal · Authors · 2025-03-31
>
> Thank you for your concrete review. Below we answer the questions and concerns you raised.
>
> > **Claims and Evidence)** There is no measurement regarding the uncertainty of the generative model or discussion about the relationships between two terms of uncertainty.
>
> While utilizing the uncertainty of the diffusion model can be an interesting future work, measuring the uncertainty of the diffusion models is mostly complex [1]. We conducted a brief literature search on this topic [2, 3], but most of them focus on detecting poor-quality images, which lies outside of our research scope.
>
> [1] Wu, Dongxia, et al. "Diff-BBO: Diffusion-Based Inverse Modeling for Black-Box Optimization."
>
> [2] Kou, Siqi, et al. "Bayesdiff: Estimating pixel-wise uncertainty in diffusion via bayesian inference."
>
> [3] Jazbec, Metod, et al. "Generative Uncertainty in Diffusion Models."
>
> > **Methods and Evaluation Criteria)** It is helpful to verify that by including the black-box optimization benchmarks with structured data in the design space, such as molecular optimization tasks.
>
> Our method can be directly applied to benchmarks with structured data, such as molecular optimization tasks. To this end, we conducted additional experiments on benchmarks with structured inputs. We follow the standard  evaluation pipeline with [4].
>
> As shown in the table, our method achieves both higher performance and better sample efficiency compared to recent BO baselines for structured inputs. We promise to add these results in our final manuscript.
>
>
> **Experiment results on structured inputs. Experiments are conducted with 4 random seeds.**
> | Tasks|# Evaluation Budget |LOL-BO [4]|CoBO [5]|DiBO|
> |:-|-|:-|:-|:-:|
> |Zaleplon MPO|20000|0.711 ± 0.014|0.724 ± 0.004|0.739 ± 0.034|
> ||30000|0.723 ± 0.006|0.728 ± 0.002|0.771 ± 0.002|
> ||40000|0.739 ± 0.000|0.738 ± 0.002|0.771 ± 0.002|
> |Perindopril MPO | 20000|0.734 ± 0.000|0.715 ± 0.025|0.815 ± 0.004|
> ||30000|0.771 ± 0.014|0.788 ± 0.024|0.818 ± 0.006|
> ||40000|0.798 ± 0.021|0.796 ± 0.018|0.825 ± 0.009|
>
> [4] Maus, Natalie, et al. "Local latent space bayesian optimization over structured inputs."
>
> [5] Lee, Seunghun, et al. "Advancing bayesian optimization via learning correlated latent space."
>
> > **Theoretical Claims)** There is no theoretical guarantee provided for the sampling approach.
>
> While we acknowledge that the theoretical guarantee of our algorithm could further enhance its reliability, the guarantee of finding optimal solutions using deep learning models is almost impossible. Our paper makes a methodological and empirical contribution to solving high-dimensional black-box optimization problems effectively. We believe that our method represents a new departure for solving high-dimensional black-box optimization by importing ideas from diffusion models and amortized posterior inference.
>
> >**Experimental Designs or Analyses)** My concern is that many baselines for high-dimensional black-box optimization are missing. For the ablation study in the appendix, it will be helpful if the authors can include other baselines in the analysis of batch size and initial dataset size.
>
> We apologize for missing some crucial baselines in high-dimensional BO. However, we would like to emphasize that we present 4 strong BO-based baselines (We also already include LA-MCTS, which you mentioned in [5]). In particular, MCMC-BO and CMA-BO, which published last year, outperform most of the baselines listed above in various benchmarks.
>
> Nevertheless, we conducted experiments with additional baselines, logEI and MCTS-VS.
> As shown in the table, we outperform those baselines in terms of performance. We promise to conduct experiments on all benchmarks and update the results in our final manuscript.
>
> **Experiment results of DiBO and additional baselines. Experiments are conducted with four random seeds.**
> || TuRBO (LogEI) | MCTS-VS-TuRBO  | DiBO (Ours) |
> |-|-|:-|:-|
> |Rastrigin|-584.09|-1089.62|-560.364|
> |HalfCheetah|-511.99|-223.175|3378.353|
>
>
> Regarding ablation studies, we also include other baselines in the analysis of batch size and initial dataset size. As shown in the table, even with different experiment settings, our method consistently outperforms other baselines by a large margin. Furthermore, as depicted in Figure 9 in the Appendix, our method does not degrade in terms of performance even with a small initial dataset size.
>
> **Ablation studies on other baselines in terms of initial experiment settings. Experiments are conducted with four random seeds.**
> ||Batch size|TuRBO|Diff-BBO|DiBO (Ours)|
> |:-|:-|:-|-|:-|
> |Rastrigin|20|-797.520|-1728.317|-573.528|
> ||50| -812.958|-1702.763|-545.124|
> ||100(Default)|-950.376|-1730.651|-560.364|
>
> ||Initial Dataset size|TuRBO|Diff-BBO|DiBO (Ours)|
> |:-|:-|-|-|:-|
> |Rastrigin|10|-1012.730|-1745.659|-586.761|
> ||50|-952.407|-1700.911|-629.776|
> ||200 (Default)|-950.376|-1730.651 |-560.364|

---

### Decision · Program_Chairs · 2025-05-01

**Decision:**

Accept (poster)

**Comment:**

This work considers a two-stage approach for black box optimization using diffusion model. The proposed method is novel and the experiments support the claims that it outperforms baselines such as BO. While no theoretical results are part of this work and better understanding the type of exploration exploitation trade-off the method performs would be of general interest (and strengthen the paper), the contributions and results are valuable to the community. The authors also provided additional convincing evidence supporting the results and claims, resolving most of the concerns raised by the reviewers.